# Free Cholesterol Accelerates Aβ Self-Assembly on Membranes at Physiological Concentration

**DOI:** 10.3390/ijms23052803

**Published:** 2022-03-03

**Authors:** Mohtadin Hashemi, Siddhartha Banerjee, Yuri L. Lyubchenko

**Affiliations:** 1Department of Pharmaceutical Sciences, University of Nebraska Medical Center, 986025 Nebraska Medical Center, Omaha, NE 68198-6025, USA; mohtadin.hashemi@unmc.edu (M.H.); sbanerjee6@ua.edu (S.B.); 2Department of Chemistry and Biochemistry, The University of Alabama, Shelby Hall, Tuscaloosa, AL 35487, USA

**Keywords:** Alzheimer’s disease, amyloid aggregation, lipid bilayer, cholesterol, time-lapse AFM imaging, molecular dynamics

## Abstract

The effects of membranes on the early-stage aggregation of amyloid β (Aβ) have come to light as potential mechanisms by which neurotoxic species are formed in Alzheimer’s disease. We have shown that direct Aβ-membrane interactions dramatically enhance the Aβ aggregation, allowing for oligomer assembly at physiologically low concentrations of the monomer. Membrane composition is also a crucial factor in this process. Our results showed that apart from phospholipids composition, cholesterol in membranes significantly enhances the aggregation kinetics. It has been reported that free cholesterol is present in plaques. Here we report that free cholesterol, along with its presence inside the membrane, further accelerate the aggregation process by producing aggregates more rapidly and of significantly larger sizes. These aggregates, which are formed on the lipid bilayer, are able to dissociate from the surface and accumulate in the bulk solution; the presence of free cholesterol accelerates this dissociation as well. All-atom molecular dynamics simulations show that cholesterol binds Aβ monomers and significantly changes the conformational sampling of Aβ monomer; more than doubling the fraction of low-energy conformations compared to those in the absence of cholesterol, which can contribute to the aggregation process. The results indicate that Aβ-lipid interaction is an important factor in the disease prone amyloid assembly process.

## 1. Introduction

The self-assembly of amyloid β (Aβ) is a process that results in the production of neurotoxic oligomer and fibrillar aggregates in Alzheimer’s disease [1,2]. Understanding the mechanism by which these aggregates are formed has been the major focus of research in Alzheimer’s disease and other fatal neurodegenerative diseases [3,4]. However, in the majority of in vitro studies, the Aβ concentrations used are several orders of magnitude higher than the physiologically relevant concentrations [5,6]; no aggregation is observed at the physiological low nanomolar concentration of Aβ. This suggests that the aggregation of Aβ in vivo utilizes pathways different from those probed by in vitro experiments.

Recently, an alternative aggregation mechanism has been discovered, allowing for the aggregation to occur at the physiologically relevant concentrations of Aβ [7,8]. This is the on-surface aggregation pathway, in which interactions with a surface act as a catalyst for the aggregation process. The model for the on-surface aggregation process suggests that the self-assembly of Aβ oligomers is initiated by the interaction of amyloid proteins with the cellular membrane. The membrane catalyzes amyloid aggregation by stabilizing an aggregation-prone conformation.

Cell membranes consist of a large variety of lipids, suggesting that numerous factors may contribute to the on-membrane aggregation of amyloids. Indeed, recent publications revealed the role of such lipids as cholesterol (Chol), sphingomyelins, and gangliosides on the formation of Aβ fibrils on membrane surfaces [9,10,11]. A very recent publication [12] demonstrated that Chol in the lipid bilayer significantly enhances the aggregation of Aβ(1-42) at nanomolar monomer concentration. Importantly, computer modeling showed that Aβ(1-42) has an elevated affinity to Chol-containing membranes, adopting a set of aggregation-prone conformations. These studies led to an aggregation model with membranes playing a critical role in triggering the aggregation process and hence, the disease state. Within this model, the membrane composition is a factor controlling the aggregation process, so a change in membrane composition can shift the ratio between monomeric and aggregated states of Aβ. This hypothesis is further strengthened by the data regarding the contribution of Chol, sphingomyelins, and gangliosides to the neurotoxicity of Aβ aggregates [13,14,15], which also highlights these lipids as prime candidates for possible disease defining parameters.

While phospholipids are the major constituent of the cellular lipid bilayer, Chol is the second most abundant lipid and provides stability to the cellular membrane. Importantly, recent findings show higher level of plasma Chol in Alzheimer’s disease patients compared to healthy controls [16]. Furthermore, Chol has been identified to be present in plaques in a 1:1 ratio with Aβ [17,18]. Other studies revealed that feeding a Chol-enriched diet to rats resulted in the enhancement of APP, Aβ, and p-tau in the cortex region, which was associated with cognitive problems [19]. In a different study, it was observed that a Chol-rich diet increased the brain Chol level and resulted in motor function impairment [20]. Furthermore, neuronal Chol content has been linked with age, with higher Chol concentration being found in mature neurons compared to younger [21]. Together these results clearly connect Chol with disease development; however, the molecular mechanism of how Chol affects disease development remains unknown.

Aggregates extracted from patient brains have revealed the existence of oligomer-lipid ensembles, pointing to possible direct interaction of free lipids with Aβ [22,23]. Additionally, recent studies [24] have reported assemblies of Aβ(1-42) monomers with Chol. These reports lead us to posit that free lipids affect the aggregation of amyloid proteins. Here we tested the hypothesis on the role of free Chol in the aggregation of Aβ, at the physiologically relevant nanomolar concentration. Time-lapse Atomic Force Microscopy (AFM) was applied to monitor the *in-situ* formation of Aβ(1-42) aggregates on supported lipid bilayers in the presence of free Chol. These studies revealed that Aβ(1-42) aggregates are formed more rapidly on the lipid bilayer in presence of free Chol. Furthermore, the aggregation kinetics of Aβ in the presence of free Chol is greatest on bilayers containing Chol. Moreover, in the presence of free Chol, aggregates accumulate more rapidly in the bulk above the membrane bilayer. Altogether, these studies revealed a critical role of free Chol on the disease-prone aggregation of Aβ(1-42), suggesting that Chol can be a trigger of the aggregation process.

## 2. Results

### 2.1. Rapid Appearance of Aggregates in Presence of Free Cholesterol

The role of free Chol in the aggregation of Aβ(1-42) was investigated on a supported lipid bilayer surface. Briefly, a mixed lipid bilayer (PC-PS), containing 1-palmitoyl-2-oleoyl-glycero-3-phosphocholine (PC) and 1-palmitoyl-2-oleoyl-sn-glycero-3-phospho-L-serine (PS), was prepared as described earlier [12]. Then, 10 nM Aβ(1-42) monomer solution with and without 100 nM Chol was deposited on the bilayer and time-lapse AFM imaging was performed to visualize the on-surface aggregation process.

Figure 1a shows the lipid bilayer surface before the addition of Aβ solution. The surface is smooth and homogeneous, with no aggregate-like features or trapped vesicles, which is critical for monitoring the on-membrane aggregation events [25,26,27]. Aggregates were detected 1 h after the addition of the Aβ solution and continued growing in numbers in the subsequent time-points of 3 h and 5 h (Figure 1b,c). To quantify the aggregation process, the volume of the aggregates, at each timepoint, was measured (Figure 1d). The plot shows that the mean aggregate volume increases as a function of incubation time on the PC-PS lipid bilayer.

As a control, we performed aggregation experiments by incubating 10 nM Aβ(1-42) on the PC-PS bilayer without Chol in solution. Comparison of the volume of aggregates formed after 5 h incubation, with and without Chol present in the solution, is shown in Figure 1e. It is evident that aggregates are significantly larger when free Chol is present in the solution during aggregation, compared to only the Aβ(1-42) in solution.

### 2.2. Acceleration of Aβ(1-42) Aggregation by Cholesterol inside Membrane

To understand if the bilayer composition is important during aggregation with free Chol in solution, we assembled a mixed bilayer with Chol, PC-PS-Chol bilayer, and followed the aggregation of Aβ in the presence of free Chol on this bilayer. Representative time-lapse AFM imaging data are shown in Figure 2 and Appendix A. Initially, the bilayer surface is smooth, Appendix A. Aggregates appear within 30 min of Aβ-Chol solution addition; a few are highlighted with white arrows in Appendix A. After 2 h of incubation, the lipid bilayer surface shows a significant number of large aggregates (Figure 2a). Quantitative volume measurements for the two time-points show the change in aggregate size (Appendix A). The aggregate size increased approximately 4 times, from ~65 nm^3^ to ~272 nm^3^, between 30 min and 2 h.

We then performed aggregation experiments with only Aβ(1-42) in solution in the presence of a PC-PS-Chol bilayer, Figure 2b. Visually it is evident that greater number of aggregates are present when free Chol is in the solution. Quantitative analysis of the two experiments shows that the volume as well as the total number of aggregates are significantly greater when Aβ(1-42) aggregates in presence of free Chol in solution, Figure 2c,d.

To validate the observations and to test whether Chol itself can form aggregate-like features on the bilayer surface, we performed time-lapse experiments on the PC-PS-Chol bilayer in presence of Chol only. Appendix A shows a large area of the bilayer surface prior to addition of Chol solution. Appendix A, shows a zoom of the same area after 2 h incubation with Chol solution. Appendix A shows another area on the bilayer surface after 2 h incubation with Chol solution; there are no aggregates or aggregate-like features on the surface of the bilayer. These observations clearly demonstrate that the aggregates, which were observed on the bilayer surface, were indeed self-assembled Aβ(1-42) oligomers and that Chol inside the membrane works in synergy with free Chol, catalyzing the self-assembly of amyloid oligomers.

### 2.3. Dynamics of Aβ(1-42) Aggregation in Presence of Free Cholesterol

After 2 h aggregation of Aβ-Chol solution on the PC-PS-Chol bilayer, the surface is practically covered with aggregates, Figure 2a. However, at 3 h significantly fewer aggregates are observed, Appendix A. While the number of aggregates become fewer with increased aggregation time, their volumes increase, Appendix A–d. Volume measurements of the aggregates after 1 h incubation show, Appendix A, that the aggregate volumes are centered around 74 nm^3^. As the aggregates become larger at 3 h, the distribution changes, and a peak around 293 nm^3^ becomes prominent. Larger aggregates also appear, Appendix A. At the 4 h incubation point the aggregates are significantly larger, with a peak around 397 nm^3^, Appendix A.

Previous studies [12] have shown that aggregates are capable of dissociating from the bilayer surface. Aggregates in the presence of free Chol show similar behavior, and the findings suggest that the presence of Chol in the solution accelerates the dissociation of aggregates. This phenomenon was tested by characterizing the accumulation of aggregates in the bulk solution above the bilayer using AFM. In these experiments, 10 nM Aβ(1-42) with 100 nM Chol solution was incubated on top of PC-PS-Chol bilayer surface. At certain time intervals an aliquot was taken from the bulk solution above the bilayer, deposited onto APS-functionalized mica, and characterized using AFM imaging. The data is assembled in Figure 3. Aggregates, accumulated in the bulk solution above the bilayer, were detected after 3 h, Figure 3a, and become more prominent after 6 h, Figure 3b. At the same time, control experiments conducted with Aβ(1-42) and Chol without the bilayer present show a negligible number of aggregates, Figure 3c. Volumes of the aggregates were also analyzed and show that the average size of the aggregates increases over time, Figure 3d. These results show that the aggregates, which dissociate from the surface, do accumulate in the bulk solution, increasing the level of soluble aggregates. The data also show that, compared with the control experiments, in which 10 nM Aβ(1-42) and 100 nM Chol were incubated without the bilayer, the presence of the bilayer leads to statistically significant more accumulation of aggregates in the bulk solution.

### 2.4. Computer Simulation of Interactions of Aβ(1-42) with Free Cholesterol

We used all-atom molecular dynamics simulations to elucidate the interaction of free Chol with Aβ(1-42) monomers. Briefly, monomeric Aβ(1-42) was placed in an explicit water box, and NaCl ions were used to neutralize the system charge and keep the ionic strength at a physiologically relevant concentration, 150 mM. Aβ(1-42) was placed at 4 nm from a single Chol molecule. Dynamics of Aβ(1-42) without Chol was simulated as a control. Five replicas of each simulation system were run for 10 μs, yielding a cumulative simulation time of 50 μs for each system.

The Aβ(1-42) monomer shows a rough free energy landscape (FEL), calculated using dihedral principle component analysis of the concatenated dataset, when in the presence of a single free Chol molecule, Figure 4a. The FEL contains well-separated energy minima in three distinct areas, two small areas to the upper and lower left, and a single, large, rough area to the right. The 10 lowest energy minima are highlighted in Figure 4a, and the representative structure for each cluster of said minima are also presented, showing the Chol molecule. These 10 clusters represent ~45.6% of the conformations sampled during the simulation. The number of protein residues in contact with Chol plotted versus the simulation time are given for each individual simulation run in Appendix A–e. It is evident that the Chol molecule does not simultaneously interact with many residues of Aβ(1-42) at any given time. In fact, the majority of interactions occur through contacts with single residues. Quantitative analysis of these data show that specific regions of Aβ(1-42) are more likely to interact with the Chol molecule, Figure 4b. The contact probability for each residue, based on the combined 50 μs dataset, shows that residues 10 through 14 are most likely to interact with Chol, followed by residues 1–8 of the N-terminal region. Residues in the central hydrophobic region (CHC, residues 17–21) are also likely interaction partners, albeit with lower probability than the aforementioned regions.

Aβ(1-42) monomer, in the absence of Chol, shows a dramatically different FEL, Figure 5, in which the deepest energy minimum is isolated and dominates by number of conformations (~11.7%) while the rest of the minima are scattered around a very rough area. Furthermore, the 10 lowest energy clusters only represent ~19.9% of the conformations sampled during the simulations. Comparing the evolution of secondary structure for the different simulations, Appendix A, shows that in both systems the Aβ(1-42) monomer is dominated by turn/bend conformations, with gradual increases in β-strand structure for each system. However, interactions with Chol seems to hinder the formation of long-lived β-strands, as in 3/5 of simulations β-strand appear and disappear more rapidly than in the control simulations without Chol, Appendix A compared to Appendix A.

## 3. Discussion

In our previous study, we have shown that the presence of Chol in the lipid bilayer facilitates aggregation of Aβ(1-42) leading to rapid formation of aggregates [12]. The number of aggregates formed in presence of Chol-containing bilayers was 6 times greater compared to the aggregates on bilayers devoid of Chol. These results revealed the critical role of Chol in the aggregation process. Here, we have shown that free Chol, in addition to Chol inside the lipid bilayer (PC-PS-Chol), has an accelerating effect on Aβ(1-42) aggregation. Results unambiguously show that free Chol can further accelerate Aβ(1-42) aggregation, as the size and number of aggregates formed in presence of free Chol are greater compared to the experiments where it is absent (Figure 1 and Figure 2). This enhanced effect of free Chol indicates the possibility of direct interaction between Chol and Aβ(1-42). Several studies have shown this type of direct binding, among them [28]. NMR studies have revealed Chol-binding regions of C99, which is the source of Aβ peptide generation due to the action of γ-secretase. The region encompassing residues 18–40 of Aβ(1-42) is observed to interact with Chol [29]. Furthermore, insertion studies of various length of peptide fragments such as Aβ(17-40), Aβ(22-35), Aβ(25-35) have shown that fragments containing residues 25–35 successfully penetrated the Chol containing monolayer [30].

The findings on direct binding of free Chol to Aβ monomers are in line with our all-atom simulations (Figure 4 and Figure 5). Moreover, the energy landscapes qualitatively support the observation of increased dynamics in the Aβ molecule in the presence of Chol (Figure 4). The presence of Chol dramatically increases the sampling of the free energy landscape, but more importantly also increases the number of sampled low-energy conformations. The 10 lowest energy minima sampled by the Aβ(1-42) monomer, in presence of Chol, make up almost 46% of total conformations sampled during the 50 μs cumulative simulations. At the same time, in the absence of Chol, the 10 lowest minima make up almost 20% of the sampled conformations. This acceleration of conformational search may be the key for how Chol affects the aggregation. Indeed, comparing interactions with membranes with and without Chol showed that the Aβ(1-42) monomer experiences a similar increased sampling when Chol is present in the membrane [12]. Additionally, the affinity of the monomer to the membrane is also changed by Chol [12,31]. Furthermore, the simulations show that dimer formation on membranes with Chol inside occur almost 2X faster than on a similar membrane without Chol [12]. The effect of Chol on the free energy and conformational sampling has also been reported for Aβ dimers and trimers [32]. In addition to significant changes to the FEL, the authors also report that presence of Chol induces greater β-structure content in the dimers and trimers of the Aβ(1-42); they also report that dimer to trimer change in β-structure is also significant when Chol is present, going from 26% to 41% [32]. The discrepancy in fraction of β-structure secondary structure between monomer and oligomers can be explained by data obtained by Ono et al., in which different pure oligomers of defined sizes were compared [33]. They reported that oligomer size has a significant effect on the structure and that there is a significant alteration of the Aβ structure going from monomer to dimer.

Our results, demonstrating the accelerating effect of free Chol on Aβ(1-42) aggregation, directly suggest that interference or blocking of Chol-Aβ interaction may suppress spontaneous self-assembly of the protein and thereby reduce the early-stage toxic oligomers. Studies following this line of thought have shown promising results. Bexarotene, which binds to the Chol-binding domain of Aβ, poses a competition for Chol towards Aβ [34,35]. Treatment with nanomolar concentration of bexarotene prevented Aβ oligomer induced Ca^2+^ flux. These data indicate that the prevention of direct interaction of Chol with Aβ can significantly reduce the toxicity caused by the oligomers [34].

One of the important findings in the present study is the increased aggregate dynamics caused by the presence of free Chol (Figure 3 and Appendix A). The data shows that, although aggregates are rapidly formed on the surface, they are not firmly attached to the bilayer and can easily leave the surface spontaneously. This hypothesis is supported by a gradual accumulation of aggregates in the bulk solution above the membrane surface (Figure 3). These data clearly show that the bilayer surface, along with the presence of free Chol, can act as a highly efficient platform for producing oligomers, which then can either participate in further aggregation or act as toxic agents. Most notable, this efficient oligomer producing process occurs at physiologically low nanomolar concentrations of Aβ(1-42).

Another aspect of the oligomers formed in the presence of free Chol is their greater size compared to those formed in the absence of free Chol. Yasumoto et al. reported that low- (LMW) and high-molecular weight (HMW) oligomers use different pathways to damage neurons, with HMW being more neurotoxic and causing more direct damage to the membranes [36]. In particular, HMW oligomers caused significantly more membrane depolarization and impaired long-term potentiation. In the context of the current study, large oligomers, produced due to interactions with free Chol, that dissociated from the membrane surface may show similar mechanism of action as the HMW oligomers tested in the aforementioned study.

Overall, the present study shows that the presence of free Chol, along with in-membrane Chol, significantly accelerates the Aβ(1-42) aggregation. This process occurs at physiologically relevant conditions, including the low nanomolar protein concentration. These findings suggests that specific lipid-Aβ interactions are critical factors for the spontaneous formation of neurotoxic oligomers. These findings further extend our model on the critical role of membrane composition in the assembly of disease-prone amyloid aggregates [12]. Our new data suggest that free Chol facilitate the aggregation process of Aβ monomers. Importantly, there is a strong synergy between the in-membrane and free Chol in this membrane mediated catalysis of Aβ aggregation at physiologically relevant conditions. Note a recent publication [37], which found accumulation of free Chol in the brain for a neurovisceral Niemann-Pick type C (NPC) disease. These findings suggest that the effects of free Chol and other lipids may also be extended to other diseases. Further neurotoxic studies of nanoaggregates assembled on the membranes, in parallel with structural characterization of such aggregates, will pave the way for the development of novel diagnostic and therapeutic strategies for AD and can be extended to other neurodegenerative diseases associated with the formation of protein deposits.

## 4. Materials and Methods

### 4.1. Materials 

Lipids were purchased from Avanti Polar Lipids, Inc. (Alabama, US). Aβ(1-42) was bought from AnaSpec (Fremont, CA, USA). Chloroform was procured from Sigma Aldrich Inc (St. Louis, MO, USA). The buffer solution that was used in this study is 20 mM HEPES, 150 mM NaCl, 10 mM CaCl_2_, pH 7.4. All other chemicals, unless otherwise specified, were procured from Sigma at analytical chemistry grade or better.

### 4.2. Preparation of Supported Lipid Bilayer 

PC-PS-Chol lipid bilayer was prepared on mica substrate as mentioned in the previous publication [12]. Briefly, POPC, POPS, and Chol vesicles were prepared by sonicating the mixture for 45 min until the mixture became clear and then deposited onto freshly cleaved mica surface attached to a glass slide. The slide was then incubated at 60 °C for 1 h. After the incubation, the sample was allowed to reach room temperature and then gently rinsed with a buffer containing 20 mM HEPES, 150 mM NaCl, pH 7.4. The bilayer was then imaged immediately by AFM in liquid.

### 4.3. Preparation of Aβ42 Protein Solution 

The method for preparing the Aβ42 stock solution was kept similar to our previous publication [12]. Briefly, lyophilized Aβ(1-42) was dissolved in 100 μL of 1,1,1,3,3,3-hexafluoroisopropanol (HFIP) at room temperature with sonication. The HFIP was then evacuated completely in a vacufuge. Anhydrous DMSO was then added to prepare the stock solution, which was then kept at −20 °C. The stock solution was diluted in the buffer solution to prepare working solutions at the necessary concentrations. Working solutions were used immediately and leftover was discarded.

### 4.4. Time-Lapse AFM Imaging

Time-lapse data were obtained using an MFP-3D instrument (Asylum Research, Santa Barbara, CA, USA). AFM imaging, in buffer medium, was carried out in tapping mode using the cantilever “E” of MSNL probes (Bruker, Santa Barbara, CA, USA). The typical resonance frequency of the cantilever in buffer was 7–9 kHz with typical spring constants of ~0.1 N/m. Scan speed was typically between 1 to 2 Hz.

At the start of each time-lapse experiment the lipid bilayer was imaged to ensure a homogenous and smooth surface, devoid of any unruptured vesicles. Aβ solution was then added, and time-lapse imaging commenced in the same area of the bilayer. The cantilever was parked after recording each frame to ensure that no damage to the lipid bilayer surface occurred due to scanning.

### 4.5. AFM Data Analysis

The presented AFM images have undergone minimal processing. Flattening was applied to the images (fitted with 1st order polynomial) with FemtoScan software (Advanced Technologies Center, Moscow, Russia). Grain analysis tool in the software was applied to measure the volume of the oligomers. The volume data were plotted as histograms using Origin Pro software (OriginLab, Northampton, MA, USA) and fitted with Gaussian distribution. The mean value of the oligomer volume for each time point was determined using the peak value of the distribution and the error bars represent the standard deviation, unless otherwise mentioned.

### 4.6. Molecular Dynamics Simulations

To investigate the interaction of Aβ(1-42) monomer with Chol, we placed an Aβ(1-42) monomer (conformation taken from [38]) at 4 nm center-of-mass (CoM) from a single Chol molecule, solvated the system in TIP3P water, neutralized with NaCl counter ions, and maintained a final NaCl concentration of 150 mM. Protein was described using the Amber ff99SB-ILDN force field [39], while Chol was described using the lipid17 force field (an extension and refinement of lipid14 [40]). A control system with only Aβ(1-42) monomer was also created in a similar manner. The systems were then energy minimized, heated to 300 K, and run for 500 ps as NVT ensemble. Production simulations were run as an NPT ensemble for 10 μs; simulations for each system were repeated five times for a total of 50 μs for each system. Simulations were performed using a 2 fs integration time step. The simulations employed periodic boundary conditions with an isotropic pressure coupling at 1 bar, a constant temperature of 300 K, non-bonded interactions truncated at 10 Å, and electrostatic interactions treated using particle-mesh Ewald [41]. Simulations were performed using the Amber18 package [42].

### 4.7. Analysis of MD Trajectories

AmberTools20 suite of programs [43], Carma [44], and VMD [45] were used to analyze the obtained simulation trajectories. Graphs and mathematical analyses were obtained using MATLAB (MathWorks, Natick, MA, USA).

## Figures and Tables

**Figure 1 ijms-23-02803-f001:**
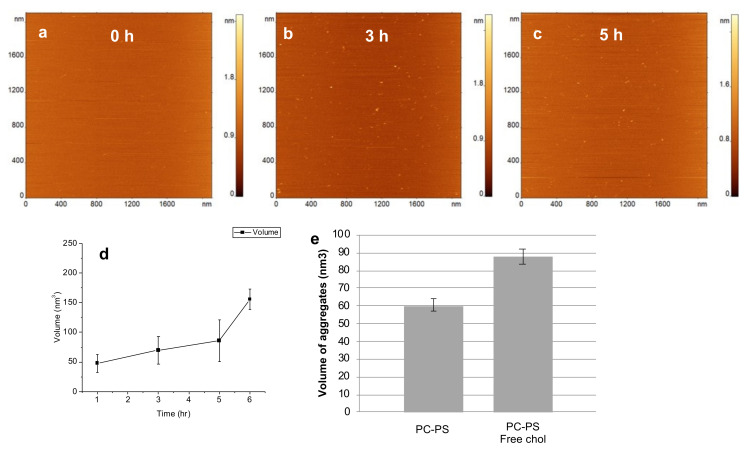
Aggregation of 10 nM Aβ(1-42), in the presence of 100 nM Chol, on PC-PS lipid bilayer. (**a**) AFM image of the bilayer surface before addition of Aβ(1-42)-Chol solution. (**b**,**c**) AFM images of the same area of the lipid bilayer 3 h and 5 h after addition of Aβ(1-42)-Chol solution. (**d**) Evolution of Aβ(1-42) aggregate volume with time. (**e**) Comparison of Aβ(1-42) aggregate volumes after 5 h incubation in the presence of PC-PS bilayer and PC-PS bilayer with Chol in solution. The volume of aggregates is significantly larger (*p* < 0.0001, *t*-test) in presence of free Chol.

**Figure 2 ijms-23-02803-f002:**
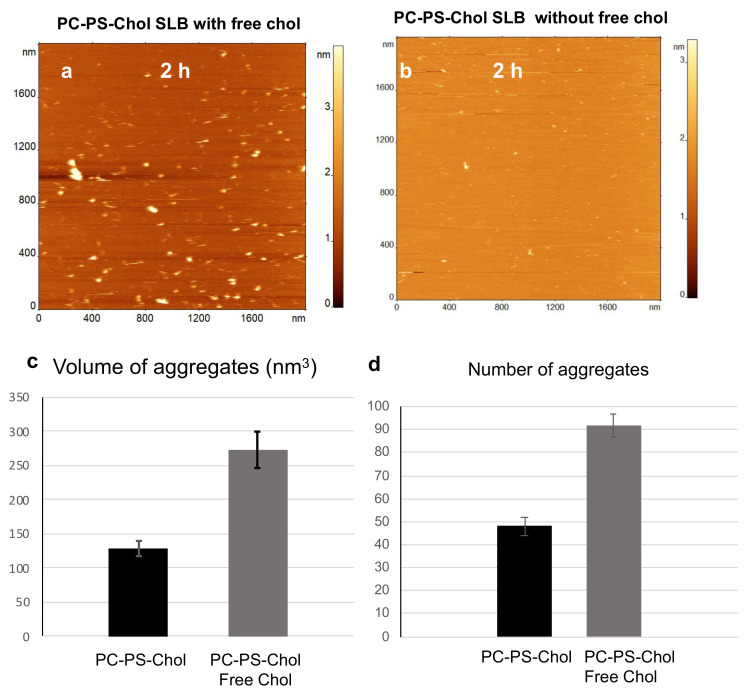
Aggregation of 10 nM Aβ(1-42) on PC-PS-Chol bilayer. (**a**) AFM image of the PC-PS-Chol lipid bilayer after 2 h incubation with 10 nM Aβ42 and 100 nM Chol in the solution. (**b**) AFM image of similar aggregation experiment as (**a**), except the absence of 100 nM Chol in the solution. (**c**) Comparison of the on-bilayer aggregate volumes in the two aggregation experiments. Data is the mean value of aggregate volumes, obtained through Gaussian fits. Presence of free Chol significantly increases (*p* = 0.001, *t*-test) oligomer volume. (**d**) Comparison of the number of aggregates formed on the lipid bilayers in the presence and absence of Chol in solution; presence of free Chol leads to significantly more oligomers (*p* = 0.003, *t*-test). For (**c**) and (**d**) the error bars represent the standard error of the mean.

**Figure 3 ijms-23-02803-f003:**
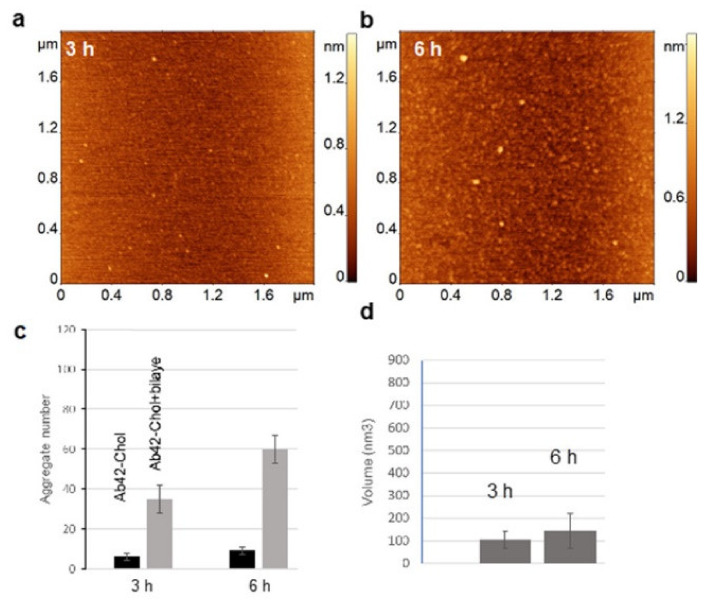
Aβ(1-42) aggregate desorption from PC-PS-Chol lipid bilayer in presence of free Chol. (**a**,**b**) AFM images of aggregates from aliquots taken from the solution above the PC-PS-Chol bilayer while 10 nM Aβ(1-42) and 100 nM Chol was incubating. Samples were taken 3 h and 6 h after addition of Aβ(1-42)-Chol solution. (**c**) Comparison of aggregates after 3 h and 6 h incubation of Aβ(1-42)-Chol in the absence and presence of PC-PS-Chol bilayer. Presence of free Chol significantly increases number of desorbed oligomers, furthermore the increase from 3 h to 6 h time point is also significant (*p* = 0.009, *t*-test). (**d**) Comparison of aggregate volumes formed in presence of free Chol, depicted in (**c**).

**Figure 4 ijms-23-02803-f004:**
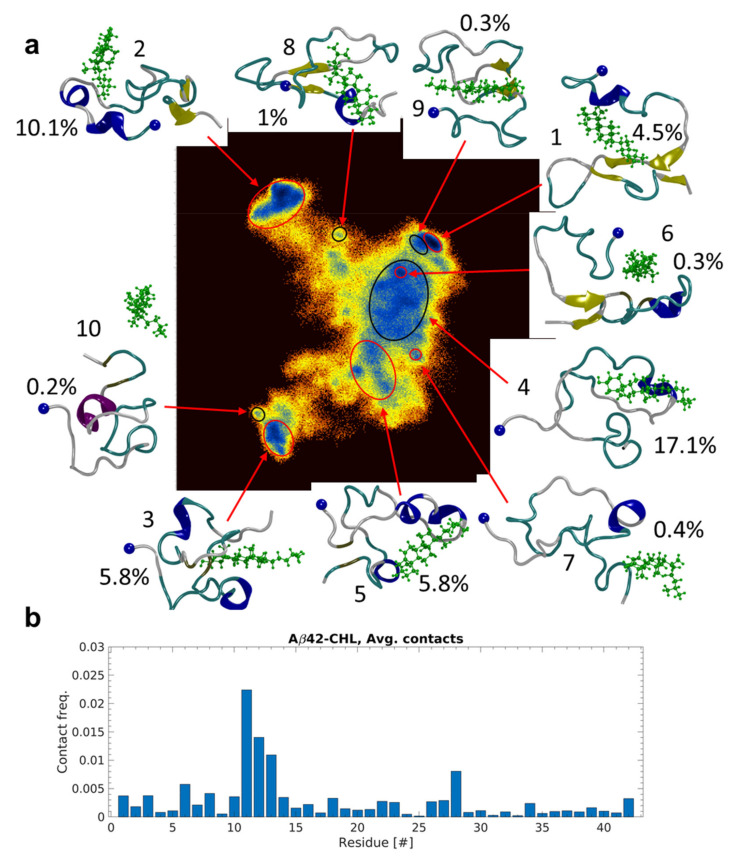
MD simulation of Aβ(1-42) interacting with Chol. (**a**) Free energy landscape based on dihedral principal component analysis of cumulative 50 μs simulation of Aβ(1-42) interacting with Chol. The 10 lowest energy minima are highlighted and the representative conformation of the Aβ(1-42) is shown. Percentages indicate the fraction of conformations relative to total number sampled during the simulations. Blue sphere denotes the N-terminal. (**b**) Average contact probability between residues of Aβ42 and the Chol molecule.

**Figure 5 ijms-23-02803-f005:**
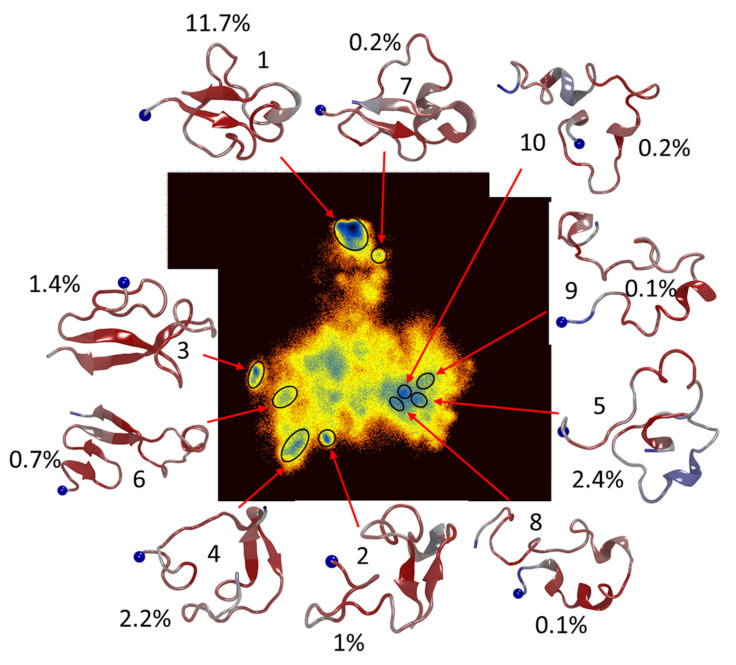
MD simulation of Aβ(1-42) monomer. Free energy landscape based on dihedral principal component analysis of cumulative 50 μs simulation of Aβ(1-42) monomer. The 10 lowest energy minima are highlighted and the representative conformation of the Aβ(1-42) is shown; colors indicate degree of fluctuation in structure, with red being highly conserved regions and blue being highly dynamic regions. Percentages indicate the fraction of conformations relative to total number sampled during the simulations. Blue sphere denotes the N-terminal.

## Data Availability

The data that support the findings of this study are available from the corresponding author upon reasonable request.

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
