# Peer review of "Free Cholesterol Accelerates Aβ Self-Assembly on Membranes at Physiological Concentration"

_ijms, 2022, doi:10.3390/ijms23052803_

Round 1
Reviewer 1 Report
Lyubchenko group showed that free phospholipid cholesterol accelerates Aβ aggregation on membranes at physiological concentrations using AFM and molecular dynamics simulations. These aggregates, which are formed on the lipid bilayer, are able to dissociate from the surface and accumulate in bulk solution; the presence of free cholesterol accelerates this dissociation as well. Molecular dynamics simulations show that cholesterol binds Aβ monomers and significantly changes the conformational sampling of Aβ monomer. It is a great and timely paper. 1. The authors should consider the interaction between Aβ and other components of membrane. 2. Apart from Aβ aggregation, does its interaction with cholesterol induce anything else on membrane in AD pathology? 3. In the section of introduction or discussion, the following related papers about the topic of Aβ oligomers and membrane damage should be included. 1) Ono K, Condon MM. Teplow DM, PNAS, 2009. 2) Ono K, Watanabe-Nakayama T. Neurochem Int, 2021. 3) Yasumoto T, Takamura Y, et al., FASEB J, 2019.Author Response
Reviewer 1
Lyubchenko group showed that free phospholipid cholesterol accelerates Aβ aggregation on membranes at physiological concentrations using AFM and molecular dynamics simulations. These aggregates, which are formed on the lipid bilayer, are able to dissociate from the surface and accumulate in bulk solution; the presence of free cholesterol accelerates this dissociation as well. Molecular dynamics simulations show that cholesterol binds Aβ monomers and significantly changes the conformational sampling of Aβ monomer. It is a great and timely paper.
We thank the reviewer for their positive evaluation of our manuscript.
- The authors should consider the interaction between Aβ and other components of membrane.
The purpose of the current paper is to unambiguously demonstrate that free cholesterol accelerates the aggregation of Ab. It is our future plan to test the effect of other free membrane lipids.
- Apart from Aβ aggregation, does its interaction with cholesterol induce anything else on membrane in AD pathology?
As described briefly in the supplemental data (Fig. S2), free cholesterol does not cause any detectable change to the bilayers under over experimental system.
- In the section of introduction or discussion, the following related papers about the topic of Aβ oligomers and membrane damage should be included. 1) Ono K, Condon MM. Teplow DM, PNAS, 2009. 2) Ono K, Watanabe-Nakayama T. Neurochem Int, 2021. 3) Yasumoto T, Takamura Y, et al., FASEB J, 2019.
We have incorporated these refs. in the paper (discussion section):
“The discrepancy in fraction of b-structure secondary structure between monomer and oligomers can be explained by data obtained by Ono et al., in which different pure oligomers of defined sizes were compared [33]. They reported that oligomer size has a significant effect on the structure and that there is a significant alteration of the Ab structure going from monomer to dimer.”
“Another aspect of the oligomers formed in the presence of free Chol is their greater size compared to those formed in the absence of free Chol. Yasumoto et al. reported that low- (LMW) and high-molecular weight (HMW) oligomers use different pathways to damage neurons, with HMW being more neurotoxic and causing more direct damage to the membranes [36]. In particular, HMW oligomers caused significantly more membrane depolarization and impaired long-term potentiation. In the context of the current study, large oligomers, produced due to interactions with free Chol, that dissociated from the membrane surface may show similar mechanism of action as the HMW oligomers tested in the aforementioned study.”
Reviewer 2 Report
The paper entitled “Free cholesterol accelerates Aβ self-assembly on membranes at physiological concentration” by Hashemi et al. deals with investigated aggregation of amyloid Aβ by cholesterol in membranes which may be a relevant factor in Alzheimer disease. The technical procedures are correct and the results are clear-cut. However, authors should revise and correct several syntax mistakes (which require major corrections):
Major concerns:
- Figure 1. what is the significance of the data? PC-PS free chol vs PC-PS
- Figure 2 C-D, Figure 3 C-D: significance and standard error of the mean are missing
- Cholesterol metabolism in the brain should be studied separately from peripheral cholesterol metabolism because it is not able to cross the Blood Brain Barrier (BBB) into the brain. However, oxidized cholesterol metabolites (oxysterols) are able to cross the BBB from the circulation. Authors should discuss it!
- At what stage in the AD pathogenesis would an Aβ-targeted therapeutic intervention show maximum efficacy? The specific lipid-Aβ interactions would be the key factors. Recently, Aducanumab, the first monoclonal antibody approved by the FDA based on reduction of the Aβ load in the brain. Please add more critical evaluation of the literature, and an impartial opinion of the pros and cons of the model used of free Cholesterol on the disease-prone aggregation of Aβ (1-42)
Typos:
Check the references [X], in several occasions they appear in italics
Author Response
Reviewer 2:
The paper entitled “Free cholesterol accelerates Aβ self-assembly on membranes at physiological concentration” by Hashemi et al. deals with investigated aggregation of amyloid Aβ by cholesterol in membranes which may be a relevant factor in Alzheimer disease. The technical procedures are correct and the results are clear-cut. However, authors should revise and correct several syntax mistakes (which require major corrections):
We thank the reviewer for their favorable evaluation of our manuscript.
Major concerns:
- Figure 1. what is the significance of the data? PC-PS free chol vs PC-PS
The information has been added to the Figure legend.
“e) Comparison of Aβ(1-42) aggregate volumes after 5 hr incubation in the presence of PC-PS bilayer and PC-PS bilayer with Chol in solution. The volume of aggregates is significantly larger (p<0.0001, t-test) in presence of free Chol.”
- Figure 2 C-D, Figure 3 C-D: significance and standard error of the mean are missing
The figures have been modified and the requested information has been added to the Figure legend.
“Fig 2: c) Comparison of the on-bilayer aggregate volumes in the two aggregation experiments. Data is the mean value of aggregate volumes, obtained through Gaussian fits. Presence of free Chol significantly increases (p=0.001, t-test) oli-gomer volume. d) Comparison of the number of aggregates formed on the lipid bilayers in the presence and absence of Chol in solution; presence of free Chol leads to significantly more oligomers (p=0.003, t-test). For c and d the error bars represent the standard error of the mean.”
“Fig 3: c) Comparison of aggregates after 3 hr and 6 hr incubation of Ab(1-42)-Chol in the absence and presence of PC-PS-Chol bilayer. Presence of free Chol significantly increases number of oligomers, furthermore the increase from 3 hr to 6 hr time point is also significant (p=0.009, t-test). d) Comparison of aggregate volumes formed in presence of free Chol, depicted in c.”
- Cholesterol metabolism in the brain should be studied separately from peripheral cholesterol metabolism because it is not able to cross the Blood Brain Barrier (BBB) into the brain. However, oxidized cholesterol metabolites (oxysterols) are able to cross the BBB from the circulation. Authors should discuss it!
The purpose of the paper is to unambiguously demonstrate that free cholesterol facilitates aggregation of Ab. It is beyond the scope of this study to determine the different pathways that may influence the presence and concentration of cholesterol or its derivative. It is in our future plans to test the effect of other molecules on the aggregation of Ab and certainly cholesterol derivatives are among those to be tested.
- At what stage in the AD pathogenesis would an Aβ-targeted therapeutic intervention show maximum efficacy? The specific lipid-Aβ interactions would be the key factors. Recently, Aducanumab, the first monoclonal antibody approved by the FDA based on reduction of the Aβ load in the brain. Please add more critical evaluation of the literature, and an impartial opinion of the pros and cons of the model used of free Cholesterol on the disease-prone aggregation of Aβ (1-42)
In our recent publications (refs 12 and 25), we discussed our model on critical role of membranes at the early stage of the disease development allowing for the self-assembly of Aβ oligomers at physiological conditions. The results of this paper suggest that free cholesterol increase further the catalytic role of membranes. We discussed briefly this effect of free cholesterol in the last paragraph of the paper. This part of the discussion section now reads as follows:
These findings further extend our model on the critical role of membrane composition in the assembly of disease-prone amyloid aggregates [12]. Our new data suggest that free Chol facilitate the aggregation process of Aβ monomers. Importantly, there is a strong synergy between the in-membrane and free Chol in this membrane mediated catalysis of Aβ aggregation at physiologically relevant conditions. Note a recent publication [37], which found accumulation of free Chol in the brain for a neurovisceral Niemann-Pick type C (NPC) disease. These findings suggest that the effects of free Chol and other lipids may also be extended to other diseases. Further neurotoxic studies of nanoaggregates assembled on the membranes, in parallel with structural characterization of such aggregates, will pave the way for the development of novel diagnostic and therapeutic strategies for AD and can be extended to other neurodegenerative diseases associated with the formation of protein deposits.
Typos:
Check the references [X], in several occasions they appear in italics
References have been checked and the paper has undergone additional English editing.
Round 2
Reviewer 2 Report
The authors have addressed all my concerns